

# The effect of multi-directional sprint training on change-of-direction speed and reactive agility of collegiate tennis players

Zhihui Zhou[1,2], Chenxi Xin[3], Yue Zhao[4] and Haijun Wu[2]

[1] Anhui Science and Technology University, Anhui, China
[2] China Institute of Sports and Health, Beijing Sport University, Beijing, China
[3] Shanghai University of Finance and Economics, Shanghai, China
[4] Sports Coaching College, Beijing Sport University, Beijing, China

Corresponding authors
Chenxi Xin, 2021241020@bsu.edu.cn
Haijun Wu,
zhouzhihui@ahstu.edu.cn

## ABSTRACT

**Objective:** The aim of this study was to evaluate the effect of random route multi-directional sprint training (MDST) compared to fixed route MDST on change-of-direction speed (CODS) and reactive agility (RA), and to investigate the correlation between CODS, RA and short-distance straight sprint speed (SDSS).
**Method:** A total of 19 collegiate tennis players from Beijing Sport University were randomly assigned to either the random route MDST group (RR group, $N = 9$, age: $22.22 \pm 2.22$ years) or the fixed route MDST group (FR group, $N = 10$, age: $21.90 \pm 1.66$ years). Both groups completed a progressive load intervention training for 3 weeks, three times a week. The RR group's random route, mirroring the distance and number of change-of-direction (COD) in the FR group's fixed route, was specifically designed. The spider run, T-drill, RA test and 5-m straight line sprint test were performed before and after the intervention.
**Results:** Both groups showed improved performance in the spider run ($p < 0.05$), T-drill ($p < 0.01$) and 5-m straight ($p < 0.001$) line sprint test after the intervention. Additionally, there was no significant difference in the improvement of CODS and SDSS between the two groups ($p > 0.05$). The RA of the RR and FR groups after the intervention was significantly higher than before intervention ($p < 0.001$), and RR group showed greater improvement in RA compared to the FR group. There was a moderate correlation between spider run and T-drill ($r = 0.523$), RA ($r = 0.388$), and no significant correlation between spider run and 5-m straight sprint ($p > 0.05$). T-drill was moderately correlated with RA ($r = 0.347$) and 5-m straight sprint ($r = 0.321$). RA was moderately correlated with 5-m straight sprint ($r = 0.551$).
**Conclusion:** Three-week multi-directional sprint training can effectively improve the change-of-direction speed, reactive agility and short-distance straight sprint speed of collegiate tennis players. And random route multi-directional sprint training has better effect on improving reactive agility.

## INTRODUCTION

Tennis is a complex sport which involves sport-specific technical skills together with a high level of numerous physical components (*Fernandez-Fernandez et al., 2009*). In tennis

---

match, player need to accelerate not only in a straight line, but also laterally and multi-directionally (*Pereira et al., 2017*; *Fernandez-Fernandez et al., 2007*), and require two to four change-of-directions (COD) per point (*Giles, Peeling & Reid, 2024*; *Li, 2018*). In other words, rapid stop and go movements together with quick COD constitute major performance determinants in tennis (*Fernandez-Fernandez, Ulbricht & Ferrauti, 2014*). Tennis players change their movement direction in response to external stimuli in tennis matches (*Munivrana, Jelaska & Tomljanović, 2022*). These COD are reactive rather than pre-planned, dedicated by external stimuli such as the ball's trajectory, the opponent's movement and other court dynamics (*O'Donoghue & Ingram, 2001*; *Cooke & Davey, 2005*). The ability to respond to external stimuli with appropriate actions or reactions during training and competitions is termed reactive agility (*Sheppard & Young, 2006*) (RA). There are studies indicated that pure COD and RA represent distinct motor abilities (*Čoh et al., 2018*; *Gabbett, Kelly & Sheppard, 2008*), with COD unable to be directly equated to RA (*Young, Dawson & Henry, 2015*), but with a completely independent ability (*Nimphius et al., 2017*).

In recent years, sports training equipment presents a trend toward the development of intelligence and diversification (*O'Reilly et al., 2018*; *Cao, 2022*). Currently, the majority of devices available on the market are capable of testing either change-of-direction speed (CODS) (*e.g.*, the T-drill test with the Smartspeed device) or conducting one of the assessment tests for RA (*e.g.*, utilizing gunfire to assess initial RA in track and field sports). However, few equipment possess the capacity for both training and evaluating CODS and RA. Following the continuous development by researchers, a recent innovation known as "Speedcourt" has emerged. This equipment is designed to enhance and assess CODS and RA, and has been demonstrated by *Düking, Born & Sperlich (2016)* to be valuable and reliable for evaluating multi-directional COD. *Born et al. (2016)* conducted a comparative analysis between the effects of Speedcourt sprint training and repeated return running with 180-degrees direction changes on the CODS and RA among elite football players. Their finding revealed that the Speedcourt training group was more effective in improving the CODS and RA. However, most studies focus on the impact of repetitive sprint training or resistance training on tennis athletes' CODS and RA (*Moya-Ramon et al., 2020*; *Sinkovic et al., 2023*). But these training methods are pre-planned and do not require additional responses to external stimuli. In this context, it is difficult to really stimulate athletes to improve their RA.

Thus, this study aims to utilize the Speedcourt (Agility training instrument V1.0) to investigate the influence of random route and fixed route multi-directional sprint training (MDST) on CODS and RA of collegiate tennis players, and analyze the interrelation between CODS, RA and short-distance straight sprint ability (SDSS).

## MATERIALS AND METHODS

### Participants

A total of 22 male collegiate tennis players from Beijing Sport University were recruited as participants, among which three participants were unable to complete the experiment due to injury and the attendance rate was not up to 95%. Notably, these injuries occurred

| Group | N | Age (years) | Height (cm) | Weight (kg) | ITN (years) | Training experience (years) |
|---|---|---|---|---|---|---|
| RR | 9 | 22.22 ± 2.22 | 177.33 ± 4.56 | 71.22 ± 3.42 | 4.44 ± 0.73 | 6.67 ± 0.62 |
| FR | 10 | 21.90 ± 1.66 | 177.50 ± 3.78 | 71.90 ± 7.09 | 4.50 ± 0.71 | 6.55 ± 0.72 |

Table 1 Basic information of subjects.

outside the experimental intervention period and were completely independent of the training regimen. The subjects were divided into two groups using a random number table: random route MDST group (RR group, $n = 9$, age: 22.22 ± 2.22 years, training experience: 6.67 ± 0.62 years) and fixed route MDST group (FR group, $n = 10$, age: 21.90 ± 1.66 years, training experience: 6.55 ± 0.72 years). Detailed information of subjects is shown in Table 1. Inclusion criteria: (1) no lower limb injury; (2) Medication and surgery history within 6 months; (3) The training experience is 6–8 years; (4) International tennis number (ITN) score must be five or above. During the experimental intervention, subjects were asked to maintain their original living habits and physical activity levels and not participate in any experiment other than this one. All subjects signed informed consent before participating in the experiment. This study was approved by the Ethics Committee of Beijing Sport University (2024202H).

## Procedures

### Testing procedure

The following testing procedures are strictly performed before and after the intervention. Prior to each test, participants completed a standardized 10-min warm-up comprising tennis-specific movements, including low-intensity running, stretching, and short sub-maximum sprints. Previous research has shown that the spider run test is effective in assessing tennis specific CODS (*Department of Youth Sports, 2012*). Therefore, participants' CODS was assessed using the spider run and T-drill tests. Moreover, participants' RA and SDSS were evaluated by a RA test and a 5-m straight line sprint test, respectively.

The spider run test was conducted in the morning session of the first day. Equipment utilized in the spider run test comprised a high-speed camera, stopwatch, tennis ball, and tennis racket. The high-speed video camera was positioned three meters behind the tennis court, while the participant stood at the midpoint of the baseline. The tennis racket was placed outside of the baseline, and the tennis ball was positioned at the intersection of the singles side lines, baseline, service line, and "T" point. Upon receiving the start command, participants retrieved the ball and returned it to the designated tennis racket in sequential order (1–5). Each participant completed three times, with the best score recorded. the test route is depicted in Fig. 1.

The T-drill test was conducted in the afternoon session of the first day. Two SmartSpeeds devices were symmetrically positioned at the midpoint of the baseline, denoted as point A, while marker buckets were placed at points B, C, and D. The midpoint of the service line represented point B, and Points C and D located at the intersection of the

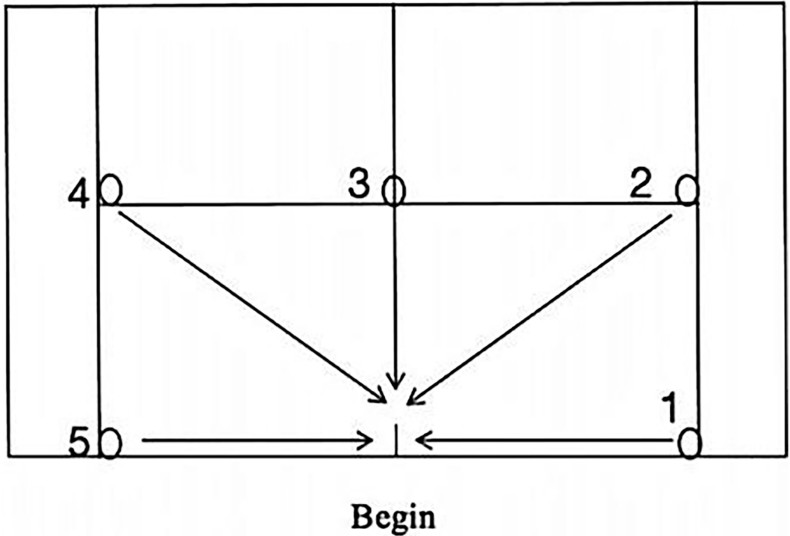

**Figure 1** Spider run test diagram.

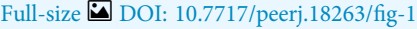

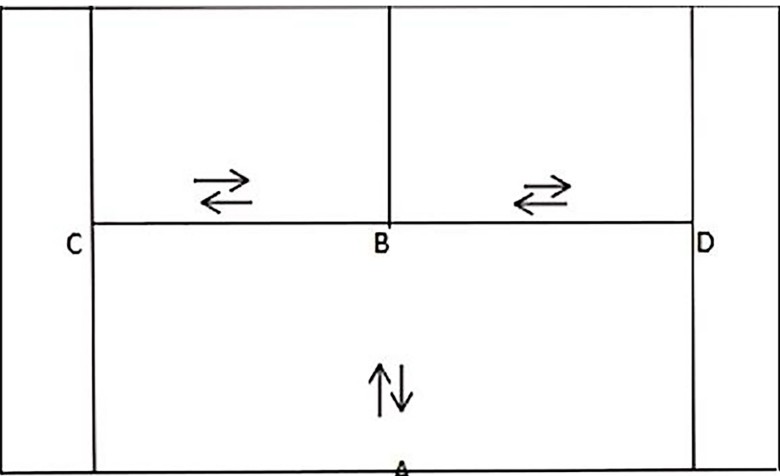

**Figure 2** T-drill test diagram.

service line's side and the singles side lines represents the touch points. The running route followed the standard "T" drill test; In this study, the distance between points A and B was approximately 5.5 m, while distances between points B and C, B and D were both approximately 4.1 m. The specific test route is illustrated in Fig. 2.

The RA test was conducted on the morning session of the next day. Identical protocol were followed for both, pre- and post-tests; however, none of the running paths utilized during either pre- or post-tests were included in any of the training sessions. In accordance with the route shown in Fig. 3, participants were instructed to complete the test three times, aiming for their best performance, with the best score recorded. Following each sprint, participants observed arrest period exceeding 30 s to ensure full recovery before commencing the subsequent sprint with full strength.
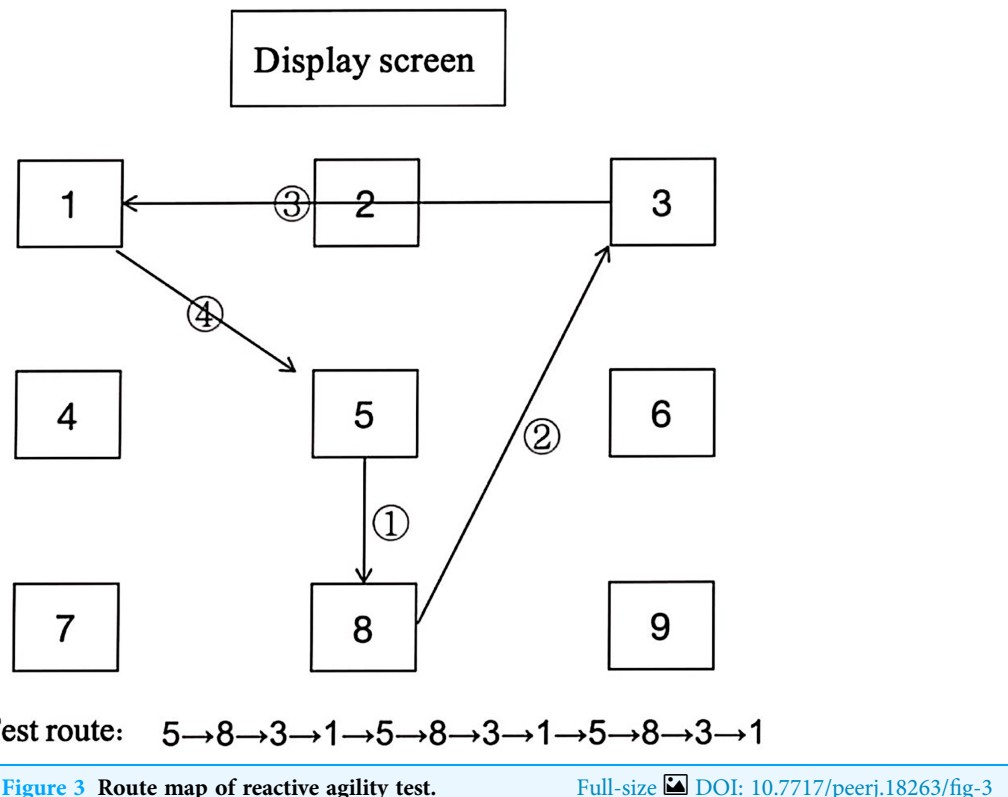

**Figure 3 Route map of reactive agility test.**

The 5-m straight line sprint test was performed on the afternoon session of the next day. Two sets of Smartspeed devices were positioned at the starting point and 5-m along the runway. Once the tester confirms the stable connection of the Smartspeed devices, the subject will be instructed to commence the sprint independently. Timing will be automatically conducted by the SmartSpeed devices through infrared detection. The best score out of three tests was recorded.

### Training procedure

The experimental intervention was conducted thrice weekly over a duration of three weeks, with a 48-h interval between each intervention. This training plan was refered to previous research (*Born et al., 2016*). Every session commenced with a standardized 10-min warm-up comprising low-intensity running, stretching, and short sub-extreme sprints, incorporating tennis-specific movements. The training intensity of both escalated weekly (First Week: 15 sessions = 3 times × 5 sets; Second Week: 20 sessions = 4 times × 5 sets; Third Week: 25 sessions = 5 times × 5 sets). A 30-s interval was observed between times, and a 5-min break was allocated between sets. Following the conclusion of the aforementioned training, participants rested for 10 min. Throughout the intervention, participants were instructed to face the trainer. The FR group employed a fixed route training regimen, with participants thoroughly acquainted with the route prior to the intervention. The screen displayed a predetermined route, with participants sprinting at maximum effort during each training session. The fixed route scheme is shown in Fig. 4. (The route design is formulated by analyzing the movement patterns observed in tennis

Warm up(10 minutes) ———→ Training ———→Warm down(10 minutes)

1st week          2nd week          3rd week
(3 times × 5 sets)   (4 times × 5 sets)   (3 times × 5 sets)

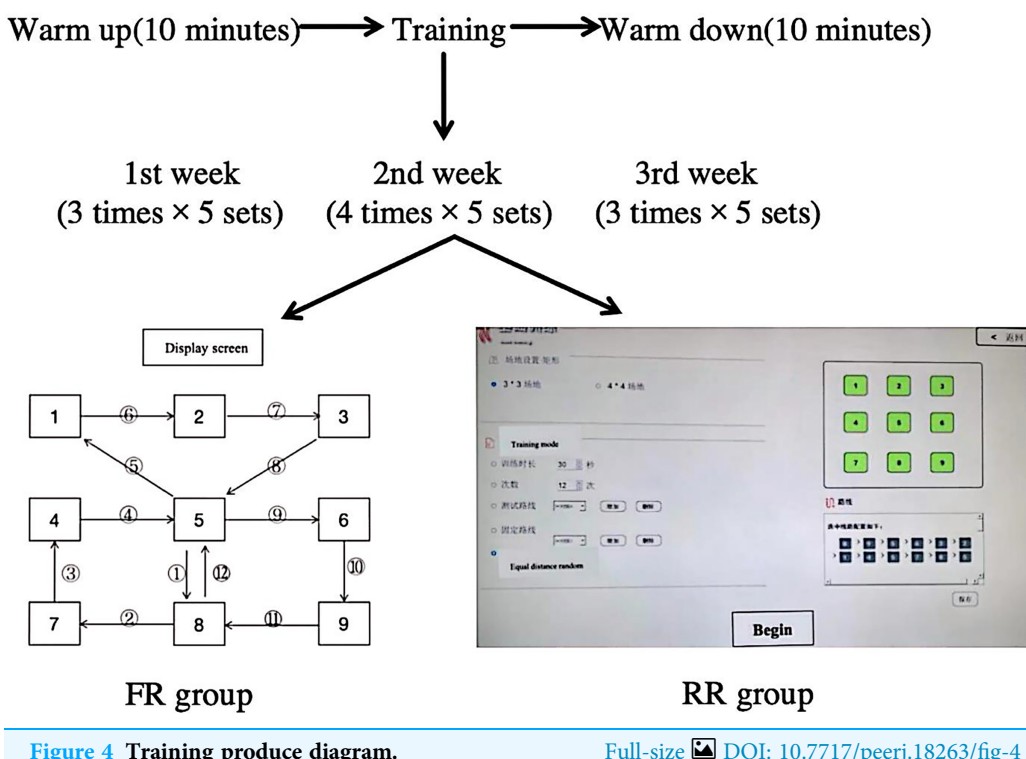

FR group                              RR group

**Figure 4 Training produce diagram.**     

matches and integrating with the capabilities of the equipment. More than 50% of the route emphasizes lateral movement, approximately 30% involves forward and backward motion, while around 20% comprises oblique upward or backward movement). The RR group utilized a random route training protocol shown in Fig. 4. The route is a random route with equal distance and equal COD designed by the Speedcourt (Agility training instrument V1.0) according to the distance of the fixed route and the number of COD. With visual signals displayed *via* the screen, upon touching each sensing point, the display initiated to release the subsequent signal. An isometric random route was established, mirroring the distance and number of COD of the fixed route. Participants were required to exert their maximal effort during each training session.

## STATISTICAL ANALYSIS

The experimental results were processed using SPSS version 19.0 statistical software. Normality of the data in each group was confirmed using the Shapiro-Wilk (S–W) test. No significant difference in baseline data was observed between two groups following independent sample t-test. Two-way ANOVA was employed to assess the intra-group data and inter-group changes before and after the intervention. In case of interaction effects group × time, simple effect analysis was conducted for each factor. Descriptive statistics was presented as "Mean ± standard deviation" (Mean ± SD). Statistical significance was set at $p < 0.05$, with high significance at $p < 0.01$. Pearson and Spearman correlation analyses were utilized to assess the relationship between related factors. Correlation coefficients (r) ranging from 0 to 0.2 indicated weak correlation or no correlation, r = 0.2–0.4 showed

**Table 2 Effect of multi-directional sprint training on CODS and 5-m straight sprint.**

| MI | Group | Pretest | Post-test | Time | | | Group | Group × Time |
|----|-------|---------|-----------|------|------|------|-------|--------------|
| | | | | F | p | η² | p | p |
| S | RR | 17.05 ± 1.25 | 16.79 ± 1.15 | 11.18 | 0.004** | 0.40 | 0.244 | 0.846 |
| | FR | 16.51 ± 0.79 | 16.28 ± 0.63 | | | | | |
| T | RR | 8.13 ± 0.29 | 7.92 ± 0.40 | 21.62 | 0.001*** | 0.56 | 0.200 | 0.728 |
| | FR | 7.88 ± 0.51 | 7.64 ± 0.51 | | | | | |
| 5 m | RR | 0.99 ± 0.09 | 0.85 ± 0.05 | 71.82 | 0.001*** | 0.81 | 0.192 | 0.957 |
| | FR | 0.97 ± 0.05 | 0.82 ± 0.03 | | | | | |

Note:
MI, measure index; S, spider run; T, T-drill; 5 m, 5-m straight sprint; η², partial η²; $p > 0.05$, there was no significant difference; **$p < 0.01$, ***$p < 0.001$ were significant.

**Table 3 Results of main effect analysis of CODS and 5-m straight sprint.**

| MI | Group | Pretest | Post-test | F | p | η² |
|----|-------|---------|-----------|---|---|-----|
| S | RR | 17.05 ± 1.25 | 16.79 ± 1.15 | 5.96 | 0.026* | 0.26 |
| | FR | 16.51 ± 0.79 | 16.28 ± 0.63 | 5.22 | 0.035* | 0.24 |
| T | RR | 8.13 ± 0.29 | 7.92 ± 0.40 | 8.77 | 0.009** | 0.34 |
| | FR | 7.88 ± 0.51 | 7.64 ± 0.51 | 13.21 | 0.002** | 0.44 |
| 5 m | RR | 0.99 ± 0.09 | 0.85 ± 0.05 | 33.67 | 0.001*** | 0.67 |
| | FR | 0.97 ± 0.05 | 0.82 ± 0.03 | 38.39 | 0.001*** | 0.69 |

Note:
MI, measure index; S, spider run; T, T-drill; 5 m, 5-m straight sprint; η², partial η²; $p > 0.05$, there was no significant difference; *$p < 0.05$, **$p < 0.01$, ***$p < 0.001$ were significant.

weak correlation, 0.2 to 0.4 indicated weak correlation, 0.4 to 0.6 indicated moderate correlation, 0.6 to 0.8 indicated strong correlation, and 0.8 to 1 indicated very strong correlation.

## RESULTS

**Effect of multi-directional sprint training on change-of-direction speed**
The data from each group were confirmed to follow a normal distribution as indicated by S–W test. The statistical analysis revealed a significant main effect of time for spider-type running (F = 11.18, $p = 0.004$, partial η² = 0.40), as depicted in Table 2. Similarly, the time main effect of T-drill measurement was notably significant (F = 21.62, $p = 0.001$, partial η² = 0.56). Moreover, there was no observed group and group × time interaction effect between the RR group and FR group for spider run ($p = 0.244$; $p = 0.846$) and T-drill ($p = 0.200$; $p = 0.728$).

Given the absence of interaction effects in both the two-factor repeated measures ANOVA of the spider run and the T-drill, the time main effect of for each was analyzed. In the spider run test (Table 3), the time main effect of the RR group was significant (F = 5.96, $p = 0.026$, partial η² = 0.26), as well as the time main effect of the FR group (F = 5.22, $p = 0.035$, partial η² = 0.24). similarly, in the T-drill test, the time main effect of RR group was significant (F = 8.77, $p = 0.009$, partial η² = 0.34), along with the time main effect of FR

**Table 4 Effect of multi-directional sprint training on RA.**

| MI | Group | Pretest | Post-test | Time | | | Group | Group × Time | | |
|----|-------|---------|-----------|------|---|---|-------|------|---|---|
| | | | | F | $p$ | $\eta2$ | $p$ | F | $p$ | $\eta^2$ |
| RA | RR | 19.16 ± 0.42 | 16.65 ± 0.30 | 125.46 | 0.001*** | 0.88 | 0.454 | 7.16 | 0.016* | 0.30 |
| | FR | 18.31 ± 0.40 | 16.77 ± 0.29 | | | | | | | |

Note:
MI, measure index; RA, reactive agility; $\eta^2$, partial $\eta^2$; $p > 0.05$, there was no significant difference; *$p < 0.05$, ***$p < 0.001$ were significant.

**Table 5 Results of simple effect analysis of RA.**

| MI | Group | Pretest | Post-test | Increase | F | $p$ | $\eta^2$ |
|----|-------|---------|-----------|----------|---|-----|------|
| RA | RR | 19.16 ± 0.42 | 16.65 ± 0.30 | 2.52 ± 0.84 | 91.46 | 0.001*** | 0.84 |
| | FR | 18.31 ± 0.40 | 16.77 ± 0.29 | 1.42 ± 0.67 | 38.36 | 0.001*** | 0.69 |

Note:
MI, measure index; RA, reactive agility; $\eta^2$, partial $\eta^2$; $p > 0.05$, there was no significant difference; ***$p < 0.001$ was significant.

group (F = 13.21, $p$ = 0.002, partial $\eta^2$ = 0.44). Notably, the difference between pre- and post-test in RR group was more significant than that in FR group in spider run test, whereas the difference in FR group was more significant than that in RR group in T-drill test.

## Effect of multi-directional sprint training on reactive agility

The Shapiro-Wilk (S–W) test confirmed that each group's data followed a normal distribution. As demonstrated in Table 4, the results of two-factor repeated measures ANOVA indicated a significant time main effect of RA test (F = 125.46, $p$ = 0.001, partial $\eta^2$ = 0.88). Additionally, there was group × time interaction effect between the RR and FR groups (F = 7.16, $p$ = 0.016, partial $\eta^2$ = 0.30).

In the RA test (Table 5), the simple effect analysis indicated the time effect of RR group was particular significant (F = 91.46, $p$ = 0.001, partial $\eta^2$ = 0.84), as well as the time simple effect of the FR group (F = 38.36, $p$ = 0.001, partial $\eta^2$ = 0.69). The RR group exhibited greater improvement in reactive agility compared to the FR (RR increase: 2.52 ± 0.84, FR increase: 1.42 ± 0.67).

## Effect of multi-directional sprint training on short-distance straight sprint speed

The Shapiro-Wilk (S–W) test confirmed that each group's data followed a normal distribution. Table 2 indicated the time main effect of in 5-m straight sprint test is significant (F = 71.82, $p$ = 0.001, partial $\eta^2$ = 0.81). Furthermore, there was no group and group × time interaction effect between the RR and FR groups ($p$ = 0.192, $p$ = 0.957).

Given the absence of an interaction effect in the two-way repeated measures ANOVA of the 5-m straight sprint test, the time main effect was analyzed. In the 5-m straight sprint test (Table 3), Both the time main effects of the RR group (F = 33.67 $p$ = 0.001, partial $\eta^2$ = 0.67) and the FR group (F = 38.39, $p$ = 0.001, partial $\eta^2$ = 0.69) were significant.

**Table 6 Correlation analysis.**

| Index | S | T | RA | 5 m |
|---|---|---|---|---|
| S | 1 | | | |
| T | 0.523* | 1 | | |
| RA | 0.388* | 0.347* | 1 | |
| 5 m | $P > 0.05$ | 0.321* | 0.551* | 1 |

Note:
   S, spider run; T, T-drill; 5 m, 5-m straight sprint; $p > 0.05$, there was no significant difference; *$p < 0.05$ was significant.

## Correlation analysis of change-of-direction speed, reactive agility and short distance straight sprint ability

Table 6 displayed the correlations between two of four variables. Spider run demonstrated moderately correlated with T-drill (r = 0.523) and RA (r = 0.388), but no significant correlation with 5-m straight sprint ($p > 0.05$). T-drill exhibited moderate correlations with RA(r = 0.347) and 5-m straight sprint (r = 0.321), while RA showed a moderate correlation with 5-m straight sprint (r = 0.551).

## DISCUSSION

Combined with the characteristics of tennis, this study investigated the influence of MST on athletes' CODS and RA, and at the same time, through the 5-m straight sprint test, the correlation between short-distance straight sprint ability and athletes' CODS and RA was investigated. The results showed that the performance of spider run, T-drill, RA and 5-m straight sprint test were improved in both groups after the intervention. Between-group comparisons showed that RA performance improved more in the RR group compared with FR. The results of correlation analysis showed that spider run was moderately correlated with T-drill and RA but not significantly correlated with 5-m straight sprint. T-drill was moderately correlated with RA and 5 m straight sprint. RA was moderately correlated with 5 m straight sprint.

In this study, spider run, which is used to evaluate athletic' CODS of athletes in the ITN test, was used as the main test index of CODS. And T-drill was used as a supplementary test index to verify CODS. The results showed that the spider run test and T-drill scores in RR and FR groups were significantly higher after the intervention than before the intervention, and there was no statistically significant difference between the two groups. A previous study (Liu, 2019) found that after eight weeks of MDST, participants showed significant improvements in measures of their CODS, namely the Illinois run, Nebraska run and T-drill. Born et al. (2016) conducted Speedcourt sprint training on 10 elite football players and observed significant improvement in the Illinois run performance after three weeks of training. A study (Sanchez-Sanchez et al., 2019) found repeated sprint with COD training improved COD performance in high-fitness players. In this study, it is considered that the CODS was increased in both groups, which may be due to the adaptation of the central nervous system to initiate the specific action program (Moya-Ramon et al., 2020; Raeder et al., 2024). However, there was no significant difference between the two groups in the improvement effect and magnitude, because there was no decision factor in these

CODS tests and the movement structure was known in advance (*Nygaard Falch, Guldteig Radergard & van den Tillaar, 2019*), so participants were less susceptible to errors during execution (*Sinkovic et al., 2023*).

The results of this study indicated that both RR and FR groups showed significant improvement in RA after intervention, and RR group showed greater improvement in RA compared with FR group. *Chaouachi et al. (2014)* observed that six weeks of MDST had a significant improvement in football players' sprint, CODS and RA. By comparing the effect of repeated return running training after three weeks with random repeated sprint training using Speedcourt training apparatus on reactive agility, it was showed that random repeated sprint training yielded better results compared to closed-loop return running training on a conventional track (*Born et al., 2016*). This study found that the RR group showed greater improvement in RA. The improvement of RA mainly depends on the improvement of perception and reaction time to a given external stimulus, rather than the actual movement speed (*Young & Rogers, 2014*). Therefore, physical training for tennis should involve multi-directional COD movement in response to external stimuli, rather than predefined COD movement. As MDST with random routes, Speedcourt can provide a valuable way to design such exercises and improve RA by providing random COD movements in all directions (*Born et al., 2016*).

The results of this study showed that the 5-m straight sprint performance of RR and FR groups was significantly better after the exercise intervention than before. Previous research (*Wang, 2022*) found that MDST was more effective in improving the performance of 5-m straight sprint compared with traditional training. The experimental results in this study are consistent with previous studies in that. The 5-m straight sprint represents a predetermined fixed route in essence, and is directly tested using a Smartspeed infrared monitor without any additional cognitive response. Thus, the random route group did not exhibit an advantage over the fixed route group in short straight-line sprints. In addition, the correlation analysis results of this study showed that the 5 m straight sprint was moderately correlated with T-drill ($r = 0.321$) and RA ($r = 0.551$). T-drill was moderately correlated with RA ($r = 0.347$) and spider run ($r = 0.523$), while spider run was moderately correlated with RA ($r = 0.388$), but not significantly correlated with 5-m straight sprint. Previous research (*Jakovljevic et al., 2012*; *Condello et al., 2013*) has reported moderate to high correlation between straight-line sprinting speed and different COD tests. This study argues that both spider run and T-drill are indicators of CODS, so their correlation with RA tends to be slightly higher. The T-drill consists of front, left, and right directions. The spider run consists of left, right, front, and oblique directions, performed by crossing or sliding steps, whereas the 5-m straight sprint requires only forward movement (*Dos'Santos et al., 2018*). Thus, it explains the lack of a significant correlation between the spider run and the 5-m straight sprint.

## CONCLUSION

Three-week multi-directional sprint training can effectively improve the change-of-direction speed, reactive agility and short-distance straight sprint speed of collegiate tennis players. Random route multi-directional sprint training has better effect on improving

reaction agility. Therefore, random multi-direction sprint training is more recommended to improve change-of-direction speed and reactive agility compared with fixed route multi-direction sprint training for tennis. In addition, Speedcourt can be used as a valuable method to design and personalize specific physical training for tennis players from a practical point of view. It is suggested that future studies could extend the training period and include more participants.

### Funding
The authors received no funding for this work.

### Competing Interests
The authors declare that they have no competing interests.

### Author Contributions
- Zhihui Zhou conceived and designed the experiments, performed the experiments, analyzed the data, prepared figures and/or tables, authored or reviewed drafts of the article, and approved the final draft.
- Chenxi Xin analyzed the data, prepared figures and/or tables, and approved the final draft.
- Yue Zhao conceived and designed the experiments, authored or reviewed drafts of the article, and approved the final draft.
- Haijun Wu conceived and designed the experiments, performed the experiments, authored or reviewed drafts of the article, and approved the final draft.

### Human Ethics
The following information was supplied relating to ethical approvals (*i.e.*, approving body and any reference numbers):

Beijing Sport University Ethics Committee.

### Data Availability
The raw measurements are available in the Supplemental File.

### Supplemental Information
Supplemental information for this article can be found online at http://dx.doi.org/10.7717/peerj.18263#supplemental-information.

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
