# Peer review of "The effect of multi-directional sprint training on change-of-direction speed and reactive agility of collegiate tennis players"

_PeerJ, doi:10.7717/peerj.18263_

## Round 0.1 · original submission · Major Revisions

Dear authors,

The study entitled “The effect of multi-direction sprint training on change-of-direction speed and reactive agility of Collegiate Tennis Players” demonstrated interesting findings using an appropriate methodological approach. However, some important points must be clarified in the manuscript. Your article has great potential for publication on PeerJ, but the reviewers have requested substantial changes to be made.

Please ensure that all review, editorial, and staff comments are addressed in a response letter and that any edits or clarifications mentioned in the letter are also inserted into the revised manuscript where appropriate.

·

Basic reporting

Thank you for the opportunity to review this manuscript, which considers some interesting, applied issues.

This study appears to be novel, and author showed an interesting point about “The effect of multi-direction sprint training on change-of direction speed and reactive agility of Collegiate Tennis Players”.

Based on what I have read, I notice a few things that would be good to correct, in order to improve the quality of the article.

Experimental design

ABSTRACT
Line 18-20 - Add the mean and standard deviation for the age of the participants for both groups.
INTRODUCTION
The introduction lacks general information about tennis and the requirements of tennis despite the mentioned speed and agility. In general, reorganize the introduction and respect its structure. The rationality of the study should be presented after reviewing the literature. It is very important that the authors mention why they decided on this particular topic and what they contribute new with this manuscript.
In lines 47-48 the reference is missing.
METHODS
Decide which word you will use throughout the text, subject or participants.
The sample of participants is relatively small, given that it is not elite. But I understand that it is not possible to add participants later.
Describe the sample of participants and the ranking of the competition in more detail. It is not enough to write that they are college players. What year of study are the participants in this study studying?
Line 97 – In addition, add the mean and standard deviation for the age of the participants for both groups.
Line 147 - Was there a minimum training attendance quota?

Validity of the findings

RESULTS
The results chapter is nicely divided into sections and easy to understand.
The only thing you need to unify to two decimal places throughout the entire text, it is clearer for the readers.

DISCUSSION
The first paragraph should state the aim of this paper and the main findings.
Generally, the discussion is too extensive and not easy to read. Reframe this section by adding mechanisms and possible reasons for the results in addition to comparing the studies. I hope that study bellow may help you to add more mechanism.

Andrašić, S., Gušić, M., Stanković, M., Mačak, D., Bradić, A., Sporiš, G., & Trajković, N. (2021). Speed, change of direction speed and reactive agility in adolescent soccer players: Age related differences. International journal of environmental research and public health, 18(11), 5883.
In this section, there are several references missing.
Line 286 – Add space

CONCLUSION
It is necessary to extend this chapter and make recommendations for further research in the future.
Figure 4 - Create a better figure for the RR group.
All the tables are quite unclear. Find a way to present them better and more clearly.
The p value should be shown in lowercase.

·

Basic reporting

Repeated errors are observed in all paragraphs of the study, including spelling mistakes, missing or extra spaces, and errors in sentence structure. Additionally, all references need to be reviewed as errors are found in most of them and the DOI is missing in all cases. Several places lack references to support statements, and some sentences are excessively long and complex, making them difficult to understand; I recommend dividing them for clarity. The flow of ideas could benefit from better organization to ensure each paragraph logically connects with the next. Use terms consistently, ensuring "change-of-direction ability" and "reactive agility" are clearly defined and differentiated. Avoid unnecessary repetition of concepts, such as the distinction between "change-of-direction ability" and "reactive agility," which is mentioned several times and could be simplified. I strongly recommend conducting a thorough review of the text to correct these issues and ensure the quality of the manuscript.

I suggest reconsidering the presentation of the values, using only two decimal places for all variables except the P value. Additionally, there is no such thing as a P=0 value; it should be modified to P<0.001. Lastly, I recommend that if there is no significant effect, the F and η² values should not be shown, as they do not provide relevant information in the absence of significance.

Experimental design

In my opinion, there are two significant aspects that limit the study. First, the sample size is inadequate for drawing robust conclusions. Second, the current duration of three weeks with nine training sessions may be insufficient for significant adaptations and optimal performance improvement. To address these obstacles, you should reference previous studies to justify that 22 participants are sufficient to achieve significant results. Generally, research indicates that longer intervention periods, often 6 to 12 weeks, are required for substantial neuromuscular and metabolic adaptations.

Additionally, there needs to be greater clarity regarding the inclusion criteria, specifying all the requirements that participants must meet. It is crucial to describe the random assignment process to the groups, detailing the method used. Furthermore, indicate whether any controls were implemented for confounding variables that could have influenced the study results.

Ensure consistency in testing protocols and justify the use of specific equipment to emphasize its importance in obtaining accurate measurements. Also, it is necessary to justify the 30-second rest period, as it seems too short in my opinion. These adjustments will help enhance the clarity and reliability of your study's findings.

Validity of the findings

The entire discussion needs to be reformulated for several reasons. First, it should clearly start with the main objectives of the study and then present the most significant findings to place the reader in the context of the research. The discussion is too lengthy and should focus on interpreting the study's findings, comparing them with previous research, and explaining their implications and limitations, without including theoretical definitions that belong in the introduction or literature review. Additionally, the results of other studies should be compared with your own findings to highlight similarities, differences, and the contribution of your work. Currently, the discussion begins with other studies without first presenting your own results, which can confuse the reader. The presentation of results is complicated and includes too many unnecessary statistical details, such as all the p-values, which can be simplified. Finally, the conclusion is presented abruptly and should be in a dedicated section or after an adequate interpretation of the results.

·

Basic reporting

No comment

Experimental design

Introduction, Methods, Discussion, Results and Conclusion are organized in a way that is easy to understand. All are presented effectively, the results are contextualized.

The research question is clear, concise and complete.

The research design is defined and clearly described and is sufficiently detailed to permit the study to be replicated.

Validity of the findings

The variables being investigated are clearly identified and presented.

Data analysis procedures are sufficiently described and are sufficiently detailed to permit the study to be replicated. Interpretations of the results are appropriate, conclusions are accurate (not misleading).

The conclusions follow from the design, methods, and results. Justification of conclusions is well articulated.

The study limitations are not discussed.

Practical significance or theoretical implications are not discussed. Guidance for future studies should be offered.

Additional comments

No additional comments

---

## Round 0.2 · accepted · Accept

Dear Author,

Congratulations! After your diligent work addressing the reviewers' comments, I am pleased to inform you that your manuscript has been accepted for publication in PeerJ. This version is more concise and formal, enhancing clarity and flow.

·

Basic reporting

No comment

Experimental design

No comment

Validity of the findings

No comment

Additional comments

No additional comments